# Neuron Activation Analysis in Multi-Joint Robot Reinforcement Learning

## Abstract

Recent experiments indicate that pre-training of end-to-end Reinforcement Learning neural networks on general tasks can speed up the training process for specific robotic applications. However, it remains open if these networks form general feature extractors and a hierarchical organization that are reused as apparent e.g. in Convolutional Neural Networks. In this paper we analyze the intrinsic neuron activation in networks trained for target reaching of robot manipulators with increasing joint number in a vertical plane. We analyze the individual neuron activity distribution in the network, introduce a pruning algorithm to reduce network size keeping the performance, and with these dense network representations we spot correlations of neuron activity patterns among networks trained for robot manipulators with different joint number. We show that the input and output network layers have more distinct neuron activation in contrast to inner layers. Our pruning algorithm reduces the network size significantly, increases the distance of neuron activation while keeping a high performance in training and evaluation. Our results demonstrate that neuron activity can be mapped among networks trained for robots with different complexity. Hereby, robots with small joint difference show higher layer-wise projection accuracy whereas more different robots mostly show projections to the first layer.

## 1 Introduction

Convolutional Neural Networks (CNN) are well known to demonstrate a strong general feature extraction capability in lower network layers. In these networks feature kernels can not only be visualized, pre-trained general feature extractors can also be reused for efficient network learning. Recent examples propose efficient reusability experimentally for Reinforcement Learning neural networks as well: Networks are pre-trained on similar tasks and continued learning for the goal application. Reusing (sub)networks that can be re-assembled for an application never seen before can reduce network training time drastically. A better understanding of uniform or inhomogeneous network structures also improves the evaluation of network performance as well unveils opportunities for the interpretability of networks which is crucial for the application of machine learning algorithms e.g. in industrial scenarios. Finally, methodologies and metrics estimating network intrinsic and inter correlations in artificial neural networks may also enhance the understanding of biological learning. Eickenberg et al. (2017) could recently demonstrate that layers serving as feature extractors in CNNs could actually be found in the Human Visual Cortex by correlating artificial networks to biological recordings.

Successful experiments to re-use end-to-end learned networks for similar tasks leave open whether such networks also self-organize feature extractors or in a dynamical domain motion primitives. Here, we analyze neuron activation in networks in order to investigate activation distribution and mapping between different networks trained on similar robot reaching tasks.

In this paper we consider a standard vertical space robot manipulator with variable number of revolute joints as the test setup for target reaching end-to-end Reinforcement Learning (RL) experiments. We introduce metrics applied to evaluate individual neuron activation over time and compare activity within individual networks all-to-all (every neuron is correlated to any other neurons in the network) and layer wise (only correlations between networks on the same layer are inspected). These metrics are utilized to set up a pruning procedure to maximize the information density in learned neural networks and reduce redundancy as well as unused network nodes. Exploiting these optimization

procedure we learn various neural networks with variable dimensions on robot manipulators with two to four joints, representing two to four Degrees of Freedom (DOF). in order to analyze similarities between network activation patterns.

As a result we demonstrate experimentally that the introduced pruning process reduces the network size efficiently keeping performance loss in bounds and hereby builds a valid basis for network analysis. We show that the networks trained and iteratively pruned on the robot manipulators form distinct neuron activation. Analyzing neuron activation correlations between different networks of various sizes, mappings between neurons trained on different manipulators can be found. A layer wise interpretation reveals that networks trained for same tasks build similar structures, but we can also discover partially similar structures between networks trained on 3 and 4 joint manipulators.

## 2 RELATED WORK

The apability of feature extraction in CNNs, alongside with a variety of analysis and visualization tools, serves as a motivation for this work on training, analysis and pruning for networks trained with RL. Analysis methods for CNNs reach from regional based methods, e.g. image occlusion Zeiler & Fergus (2014), that aim to expose the region of an image most relevant for classification, to feature based methods, e.g. deconvolution Zeiler & Fergus (2014) or guided backpropagation Selvaraju et al. (2017). Methods combining the described techniques are for example introduced as Grad-CAM in Springenberg et al. (2014). These networks demonstrate class discrimination for features of deeper network layers (Zeiler & Fergus (2014)) as a basis to apply such general feature extractors to different applications after pre-training. Pre-trained networks such as ResNet He et al. (2016), which has been trained on the ImageNet1 data set, speed up training drastically by initializing CNNs applied for similar tasks. Kopuklu2019 demonstrated that even reusing individual layers in the same network can lead to a performance increase.

Recent advances pushed RL agents to reach super human performance in playing Atari video games Bellemare et al. (2013) Mnih et al. (2015), Chess Silver et al. (2017) and Go Silver et al. (2016). These results were extended to cope with continuous action spaces in e.g. Lillicrap et al. (2015) and demonstrated great performance on highly dynamic multi-actuated locomotion learning tasks such as demonstrated in the NIPS 2017 Learning to Run challenge Kidziński et al. (2018). Vuong et al. (2019) and Eramo et al. (2020) demonstrate experimentally that knowledge learned by a neural network can be reused for other tasks in order to speed up training and hereby translate modularity concepts from CNNs to RL frameworks. Hierarchical Reinforcement Learning incorporates these ideas, utilizing the concept of subtask solving into neural networks e.g. in Andreas et al. (2016) for question answering. A successful example of transfer learning to build up a general knowledge base could be demonstrated with RL in Atari games in Parisotto et al. (2016). Gaier & Ha (2019) emphasizes the importance of neural architectures that can perform well even without weight learning.

With a main motivation to improve learning efficiency and reduce computational requirements, network pruning is introduced for various network architectures. Early work in LeCun et al. (1990) utilizes second derivative information as a heuristic to decrease network size, recent work in Livne & Cohen (2020) introduces network pruning for Deep Reinforcement Learning based on redundancy detection in an iterative process.

Li et al. (2018)

## 3 EXPERIMENTAL SETUP

In this paper we focus on a robot manipulator with operation limited to a vertical plane. A neural network is trained with end-to-end Reinforcement Learning in order to reach predefined locations in 2D space without prior knowledge of neither robot dynamics nor the environment. Hereby, end-to-end refers to a mapping from sensory feedback in terms actual joint positions in cartesian space and the desired goal location to output actions as joint position commands. We apply Deep q-learning, as proposed in Mnih et al. (2015), to predict q-values, an action is selected by means to the softmax exploration policy and Gradient descent of the networks weights is handled by the Adam Solver Kingma & Ba (2014).

For performance reasons our experiments are executed within a simplified simulation environment as shown conceptually in Figure 1 (right), but exemplary behaviors have been successfully trans-

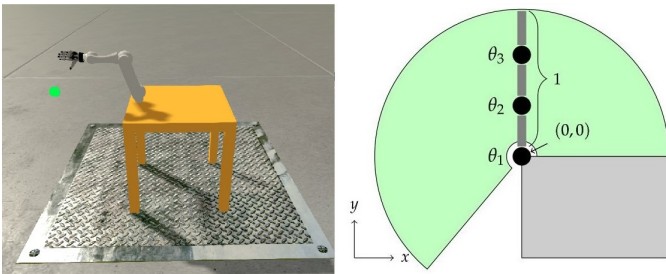

Figure 1: Neuron activity is analyzed in neural networks trained for target reaching of multi-joint robotic manipulators utilizing end-to-end Deep Q-Learning. We train the network in a simplified environment of a robot with 2 to 4 controllable joints operating in vertical space (right, initial configuration with joints $\theta_i$). Transferability to a sophisticated robotic simulation (left) with motions restricted to vertical space and the robot finger tips supposed to reach the green sphere could be demonstrated.

ferred to a realistic robotic simulation (Figure 1 left). We simulate robots with 2 to 4 DOF that are implemented as revolute joints restricted to vertical space motions and actuated with PID position controllers. For all experiments, the neural networks originally consist of 6 fully connected hidden layers with ReLU activation functions, but may be reduced in the pruning process we introduce. The network input vector $\boldsymbol{x}$ encodes actual robot joint angles $\hat{\theta}_i$ as their sine and cosine contribution for every control step $t$ (control cycle time of 50ms) as well as the desired goal position in cartesian coordinates $[x^*, y^*]$ as

$$\boldsymbol{x}^{(t)} = \begin{bmatrix} sin\left(\hat{\theta}_1^{(t)}\right) & cos\left(\hat{\theta}_1^{(t)}\right) & ... & sin\left(\hat{\theta}_n^{(t)}\right) & cos\left(\hat{\theta}_n^{(t)}\right) & x^* & y^* \end{bmatrix}^T. \tag{1}$$

The output layer contains $3^n$ neurons as the action of every individual joint $i$ is quantized into the three change of motion states $\{1, -1, 0\}$ as forward, backward and no motion for each joint with joint angle changes of $\pm 0.05 rad$. The goal state of an agent is a randomly instantiated 2D location to be reached with the robot finger tip in max. 60 control steps, each representing $50ms$. The distance between the goal position $p^*$ and the tip $\hat{p}$ is mapped into $[0, 1]$ and squared to serve as the reward function $r\left(t_i\right) := \left(\frac{1}{|\hat{p}(t_i) - p^*|_{L2} + 1}\right)^2$. All network results that are presented passed a validation test consisting of 300 test episodes. This test also serves as the pruning baseline: The probability of a type two error for reaching the final reward threshold $\bar{r} = 0.9$ with an accuracy $\bar{\rho} = 0.9$ lies bellow significance $\alpha = 0.05$ on the test data.

## 4 NEURON ACTIVATION ANALYSIS

We first analyze individual neuron activation inside multiple neural networks trained on the introduced target reaching robotic manipulator. This initial analysis servers as a baseline for pruning and projection evaluation, therefore we study only 3 joint robotic manipulators in depth before we investigate a comparison of different kinematic structures.

We define a distance metric between neurons that is based on the neuron activation history in scope of every episode in order to account for the dynamics in motion trajectory learning. All neuron activation values over the course of an episode are collected in a vector $\boldsymbol{z}_{n_i}^{(E)}$ for every neuron $n_i$ of the network in Episode $E$. Utilizing the linearity of applied ReLU activation functions we normalize this activation in range $[0, 1]$ in reference to the maximum value attained. For a set of sample episodes $\mathcal{E}$, representing a set of potential robot actions, we define the distance of neurons $n_i$ and $n_j$ as

$$d(n_i, n_j) := \frac{1}{|\mathcal{E}|} \sum_{E \in \mathcal{E}} \left| \frac{\boldsymbol{z}_{n_i}^{(E)}}{Z_{n_i}} - \frac{\boldsymbol{z}_{n_j}^{(E)}}{Z_{n_j}} \right|_{L2}, \tag{2}$$

with $\boldsymbol{z}_{n_i}^{(E)} \in \mathbb{R}_{\geq 0}^T$ denoting the vector containing activation series of neuron $n_i$ in episode $E$ and $Z_{n_i} \in \mathbb{R}_{>0}$ the maximum activation of $n_i$ in all episodes $\mathcal{E}$. For a layer wise analysis Equation 2 is adapted accordingly, only considering distances to neurons that belong to the same layer. The

upper triangular matrix of a distance matrix $D$ holds all values $d(n_i, n_j)$ with $i? = j$. The density distribution of neuron distances can be approximated by collecting all values in the upper triangular matrices of D.

Additionally, hierarchical clustering as described in Hastie et al. (2009) is applied to individual network layers in order to reveal neuron groups that show similar activation behavior. We form groups that minimize the mean cluster distance $D(C_l)$ of contained neurons as

$$D(C_l) := \frac{1}{|C_l|\,(|C_l| - 1)} \sum_{n_{il} \in C_l} \sum_{n_{jl} \in C_l \setminus \{n_{il}\}} d(n_{il}, n_{jl}). \tag{3}$$

for neuron cluster $C$ of layer $l$. We conduct an experiment with a set of $M = 20$ networks (48 neurons per hidden layer), for the three joint manipulation task. A reference set of untrained networks with identical structure is initialized by Xavier initialization Glorot & Bengio (2010). Neuron distances are averaged from a set of $m = 500$ sample episodes. The distance distribution in randomized networks forms a bell-shaped distribution globally as well as layer wise (Figure 2, top). However, the all-to-all distribution of trained networks primarily indicate a lower standard deviation and mean compared to random networks, with a slight distortion at high distances. Layer-wise analysis reveals that these higher distance scores occur increasingly on network layers closer to the output, in particular in the second half of layers. In contrast, lower layers demonstrate close to normal distributions. Clustering reveals a variety in distances for all layers in untrained randomly initialized networks (Figure 2 bottom) which is kept on the first layer only in trained networks. In particular on the middle layers clusters with low distances emerge during training.

The intrinsic network analysis depicts successful training that visibly changes the neuron activation characteristics which highly depends on the location inside the network.

## 5 HEURISTIC NETWORK PRUNING

Non-uniform density distributions and low cluster differences in the inspected neuron activation indicate potential for network pruning. Dense information representation is a requirement for the comparison of different networks. For this purpose we propose a pruning procedure that iteratively unifies neurons with similar activation, identified as small cluster distances, and retrains the network. Hereby a trade-off between reduced network size and maintaining high-performance learning is aspired.

We apply Breadth First Search on the resulting cluster tree of every network layer. The first encountered clusters with distance (3) below threshold $\bar{d}_\tau$, which is defined as a fraction $\tau$ of the maximum cluster distance, are selected to form the layer segmentation $\mathcal{C}$. Based on this neuron segmentation $\mathcal{C}^{(l)}$ of layer $l$, a reduced network is constructed that represents every cluster as a single neuron. Original network weights are reused for the initialization of the reduced network. We exploit the linearity of ReLU activation functions and assume identical neuron behavior only altered by linear scaling inside every cluster. W.l.o.g. cluster activation $\zeta_C$ are defined such that scaling factors $\gamma_n > 0$ of contained neurons sum to one and $\forall n \in C : \zeta_C = \frac{z_n}{\gamma_n}$, with $z_n$ denoting the activation of neuron $n$, holds. For cluster $C \in \mathcal{C}^{(l)}$ and arbitrary neuron $n \in C$ the forward propagation of $z_n$ can be rearranged to form the forward propagation of the cluster activation as

$$\zeta_C = ReLU \left[ \sum_{D \in \mathcal{C}^{(l-1)}} \zeta_D \frac{1}{\gamma_n} \sum_{m \in D} w_{nm} \gamma_m \right], \tag{4}$$

with $w_{nm}$ denoting the weight from neuron $m$ to $n$. (4) acts as an approximation that in practice is only achieved by clusters of dead neurons that are not activated at all. Therefore, in order to improve stability all neurons of a cluster contribute to the reduced network weights $\omega$ as $\omega_{CD} = \frac{1}{\gamma_n} \sum_{m \in D} w_{nm} \gamma_m$. Scaling factors $\gamma_n$ are generated from the maximum activation $Z_n$ (2) of the respective neuron $n$.

In order to evaluate the introduced pruning procedure, we conduct experiments with a set of $M = 20$ neural networks (6 hidden layers, 48 neurons each) trained for the 3 joint manipulation task. Network reduction is applied with a set of $m = 300$ sample episodes, presented results are averaged over the set of networks which reached sufficient performance. The results presented in Figure 3 (left) show a nearly linear correlation between cluster threshold and resulting pruned network size if networks had an identical initial layer size of 48 neurons. In case of $\tau = 0$ only dead neurons

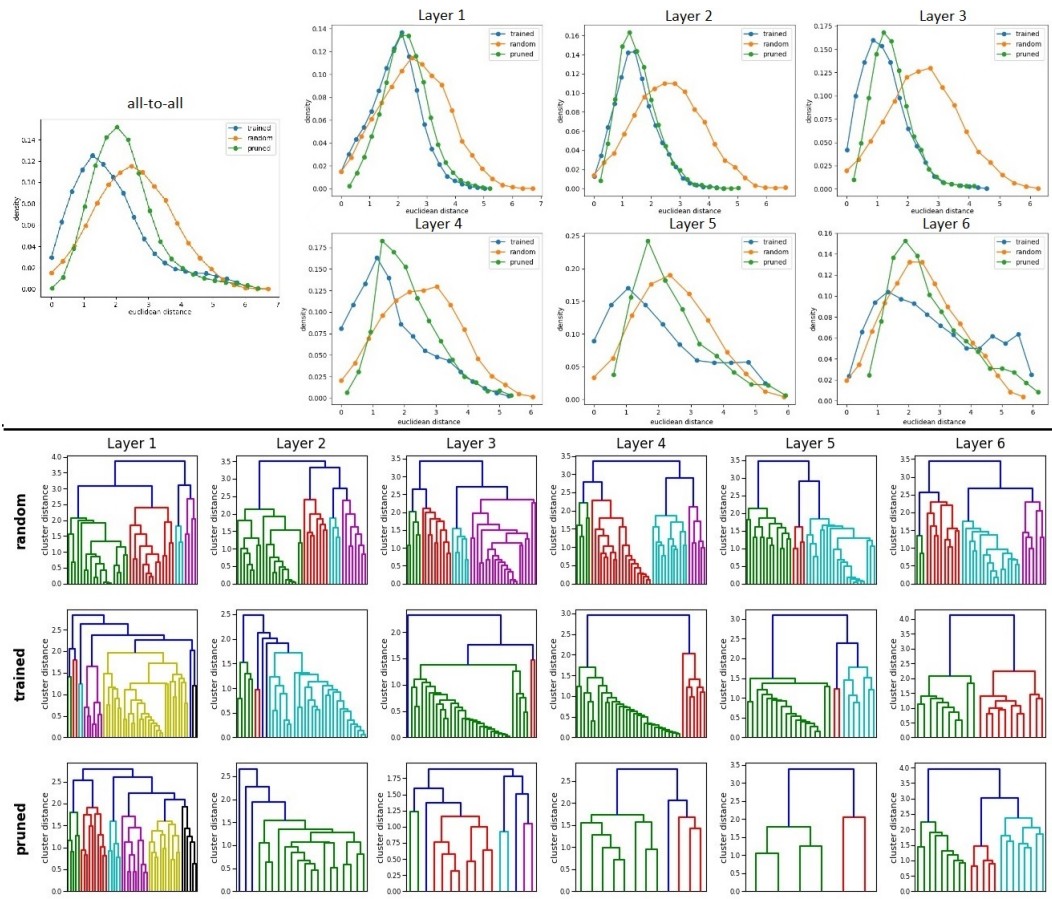

Figure 2: Neuron activation analysis for randomly initialized, trained and pruned networks on a three joint manipulator (averages over 500 sample episodes, 20 trained agents). **Top:** Distance measured between all-to-all (all neurons in a network are correlated among each other, left) neurons and layer-wise (for every neuron only neurons on this layer are considered for correlation, right) indicate a bell-shaped distribution with higher mean in the first and last layer. Pruning sharpens the bell-shape, increasing the mean, but reducing very high distance scores. **Bottom:** Clustering Dendrograms are generated based on the distance measures for an exemplary trained network. Untrained networks show very similar clusters, trained network highlights cluster groups and pruning reduces neurons while increasing cluster distance. The first layer generally keeps the most distinct clusters, the penultimate layer the strongest neuron reduction.

are reduced, which does not affect the performance of the network, though reduces the network size significantly (initial size of all networks: 323 neurons). For values of $\tau \in (0, 0.1]$ the network is reduced, but no strong effect is apparent on initial accuracy [%] and training duration (number of episodes executed until the validation set is passed). We observe interesting behavior in range of $\tau \in (0.1, 0.22]$, as the initial accuracy decreases significantly, whereas the duration for retraining the networks barely increases. This implicates that the main processes in the network remain in tact after reduction, whereas for $\tau > 0.2$ a strong increase in training duration indicates a loss of relevant information. As a trade off between minimal network size and efficient training $\tau = 0.2$ has been selected as the optimal cluster threshold and was applied for all further experiments. As the pruning process highly depends on the initial network size, we analyze networks of initial hidden layer sizes of 32, 48, 128, 256 and 512 within the same test setup. The results shown in Figure 3 (right) emphasize the first reduction step as the most dominant. Noticeably large networks of initial layer neuron count of 128, 256 and 512 reach similar pruned network size already in the first iteration step. For subsequent reduction steps the network size plateaus. Inspection of neuron per layer counts reveal that small initial networks (32, 48) taper with depth, compared to bigger initial networks that form an hourglass shape. The average network shape of 256 and 512 neuron networks after three

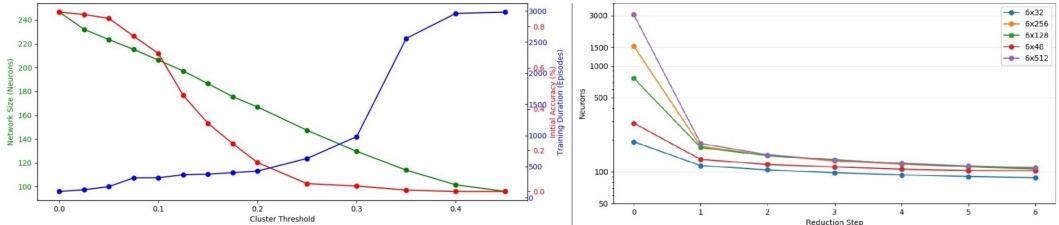

Figure 3: Evaluation of heuristic cluster based network pruning on the example of a three joint manipulator. **Left**: Even though the initial network accuracy decreases rapidly, the training duration (Number of episodes executed until the validation set is passed) only increases significantly with cluster thresholds larger than 0.3. As a trade off between minimal network size after pruning and efficient training $\tau = 0.2$ has been picked as the optimal cluster threshold and was applied for all further experiments. **Right**: The first reduction step demonstrates the strongest reduction for all networks initialized with different initial neuron count per layer. Layers with more than 128 neurons are reduced to a very similar neuron count in the first pruning step.

reduction steps turns out as $\bar{s} = [51.6 \quad 21. \quad 15.9 \quad 12.6 \quad 10.2 \quad 16.7]$ Network intrinsic neuron distance densities of pruned networks (Figure 2) implicate an increased homogeneous information representation compared to networks trained straight away. The bell-shaped distribution with higher mean shows lower variance, and outliers of high distance scores are reduced. While clusters remain rather similar on the first and last layer, in particular the cluster distances on middle layers are drastically increased along with the reduced cluster number. Overall, we find that our pruning process reduces network size efficiently and hereby shows a visible effect on neuron activation towards a rather uniform distribution and distinct cluster structure.

## 6 Correlations in Networks trained for Multi-Joint Robots

Based on the both the individual neuron activation analysis and heuristic network pruning, we now investigate mappings of neuron activation between different networks learned on robot manipulators with 2 to 4 joints. Here, the goal is to estimate whether activation patterns are similar in networks trained for the different robot kinematics. For this purpose we construct an unidirectional linear projection between source and target network and analyze its accuracy and structure. Based on the source network neuron activation $\boldsymbol{b} \in \mathbb{R}^{K}_{\geq 0}$, resulting from input $\boldsymbol{x}$, a prediction $\hat{\boldsymbol{a}} = \boldsymbol{b}^T P$ of the target activation $\boldsymbol{a} \in \mathbb{R}^{M}_{\geq 0}$ for the same input $\boldsymbol{x}$ is given by projection matrix $P \in \mathbb{R}^{K \times M}_{\geq 0}$ (Figure 4). The projection is constructed based on a set of $N$ training inputs $X$ that yield activation matrices

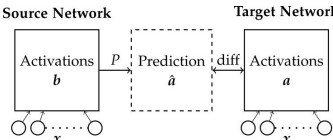

Figure 4: Analysis of inter network mappings: Sets of two networks are trained on robots with different number of joints. A projection matrix $P$ that reflects the network similarity is calculated to compute $\hat{\boldsymbol{a}}$ from the source network neuron activation $\boldsymbol{b}$ with minimal difference to $\boldsymbol{a}$.

$A \in \mathbb{R}^{N \times M}_{\geq 0}$ and $B \in \mathbb{R}^{N \times K}_{\geq 0}$ of the target and source network, respectively. In order to obtain a procedure invariant to neuron scaling, individual columns of $A$ and $B$, are normalized to the interval $[0, 1]$ dividing by the maximal values contained. The resulting projection $\bar{P}$ can be adjusted to fit the original training data by $P_{km} = \frac{\alpha_m}{\beta_k} \bar{P}_{km}$. Two approaches for projection construction are considered. `Greedy mapping` predicts each target neuron from the source neuron with minimal distance (2), every entry of the greedy projection matrix $\bar{P}^g_{km}$ is 1 if $k = \arg\min_{i \in [K]}\{d(m, i)\}$ and 0 otherwise. `Linear mapping` incorporates all source neurons into the prediction of a target neuron by linear combination. Projection vectors $\boldsymbol{p}_m$, predicting the behavior of neuron $m$, are given by the solution of quadratic optimization with linear boundary constraints for each target neuron individually. Hereby, the mean squared error plus lasso regularization, to enforce sparsity of solution vectors, is minimized finding the best projection $\boldsymbol{p}$, i.e.

$$\text{minimize} \quad \frac{1}{2}\left|\bar{B}\boldsymbol{p} - \frac{\boldsymbol{a}^{\downarrow}_m}{\alpha_m}\right|^2_{L2} + \lambda|\boldsymbol{p}|_{L1} \quad \text{subject to} \quad \boldsymbol{p} \geq \boldsymbol{0}. \tag{5}$$

$\bar{B}$ denotes the matrix of source activations scaled by $\beta_k$, $\boldsymbol{a}_m$ the target activations and $\lambda \in \mathbb{R}_{\geq 0}$ the regularization strength. As mapping of two networks should be invariant to neuron scaling, all individual neuron activations are projected into the interval [0, 1] with neuron specific scaling factors $\beta_k$ and $\alpha_m$ for the source and target network neurons, respectively. 30 The solution vectors $\bar{\boldsymbol{p}}_m^*$ are stacked to form the linear projection matrix $\bar{P}^l := [\bar{\boldsymbol{p}}_1^* \quad ... \quad \bar{\boldsymbol{p}}_M^*]$. Input samples $X$ are deduced from a set of sample episodes of the target network without duplicates. In put vectors of robot manipulators with different joint count are transformed by either duplicating best aligning joints or unspecified joints being set to zero, for a more or less complex source network, respectively (Figure 5 middle right).

## 6.1 EVALUATION METRICS

Projections are evaluated with regard to their goodness to fit a set of validation samples $X^{\mathcal{V}}$ and according to heuristic metrics that directly analyze a projection structure. The mean absolute prediction error is normalized by the prediction error of the zero projection $P_0 \in \{0\}^{K \times M}$ to construct the normalized error $E(P, X)$ that is invariant to weight scaling and adding dead neurons:

$$\bar{E}(P, X) := \frac{E(P, X)}{E(P_0, X)} = \frac{1}{|A|_1} \sum_{m=1}^{M} |\boldsymbol{a}_m^{\downarrow} - B\boldsymbol{p}_m|_1 \qquad (6)$$

The entropy of a target neuron's projection $\boldsymbol{p}_m$ is referred to as the saturation of neuron $m$, projection $P$ is the mean of all neuron saturations. A low saturation implies that few neurons suffice to describe the behavior of $m$. We calculate the overall projection saturation $S(P)$ according to Equation 7.

$$\mathcal{S}(P) := -\frac{1}{M} \sum_{m=1}^{M} \sum_{k=1}^{K} P_{km} \log_K(P_{km}) \in [0, 1]. \qquad (7)$$

The utilization of the source network neurons to describe the target network is indicated by the coverage $\mathcal{C}$. It is defined as the entropy of the stochastic process that picks a target neuron $m$ uniformly at random and passes it on to the source network according to the distribution $\frac{\boldsymbol{p}_m}{|\boldsymbol{p}_m|_{L1}}$. A low coverage value implies low utilization of the source network.

$$\mathcal{C}(P) := -\frac{1}{K} \sum_{k=1}^{K} \kappa_k \log_K(\kappa_k), \quad \text{with } \kappa_k = \frac{1}{M} \sum_{m=1}^{M} \frac{P_{km}}{|\boldsymbol{p}_m|_{L1}}. \qquad (8)$$

The same statistical process is applied to construct a layer-wise projection $\mathcal{P}_{ij}$. It describes the probability of reaching the $i$th layer $L_i^{(K)}$ of the source network when starting in some uniformly random neuron in the $j$th layer $L_j^{(M)}$ of the target network.

$$\mathcal{P}_{ij} := \frac{1}{|L_j^{(M)}|} \sum_{k \in L_i^{(K)}} \sum_{m \in L_j^{(M)}} \frac{P_{km}}{|\boldsymbol{p}_m|_{L1}}. \qquad (9)$$

## 6.2 RESULTS

For each robot manipulator with 2, 3 and 4 joints, $M = 5$ networks are trained, pruned in three steps and we analyze all possible mappings "a-b" between the respective sets. A set of validation inputs $X^{\mathcal{V}}$, is generated for $m = 300$ sample episodes of the target network and metrics evaluated. As a baseline we map all 3 joint manipulator agent networks with an initial neuron count of 256 for each of the 6 hidden fully connected layers, among each other. As expected, as a baseline mappings of networks to themselves (referred to as reflexive mapping) show zero error and saturation and coverage of 1 (Figure 5 top left). However, greedy mapping shows a high normalized error and low coverage when compared to the linear mapping and thus is considered an inferior approach. In this baseline we extract linear mapping with regularization strength of $\lambda = 50$ as the best metric as it indicates coverage and normalized error most significant on trained in contrast to random networks. Layer-wise linear projection ($\lambda = 50$) is not optimal but we observe the best mapping to the respective layers, shown on the diagonal axis in the table of Figure 5. Hereby, layer one and

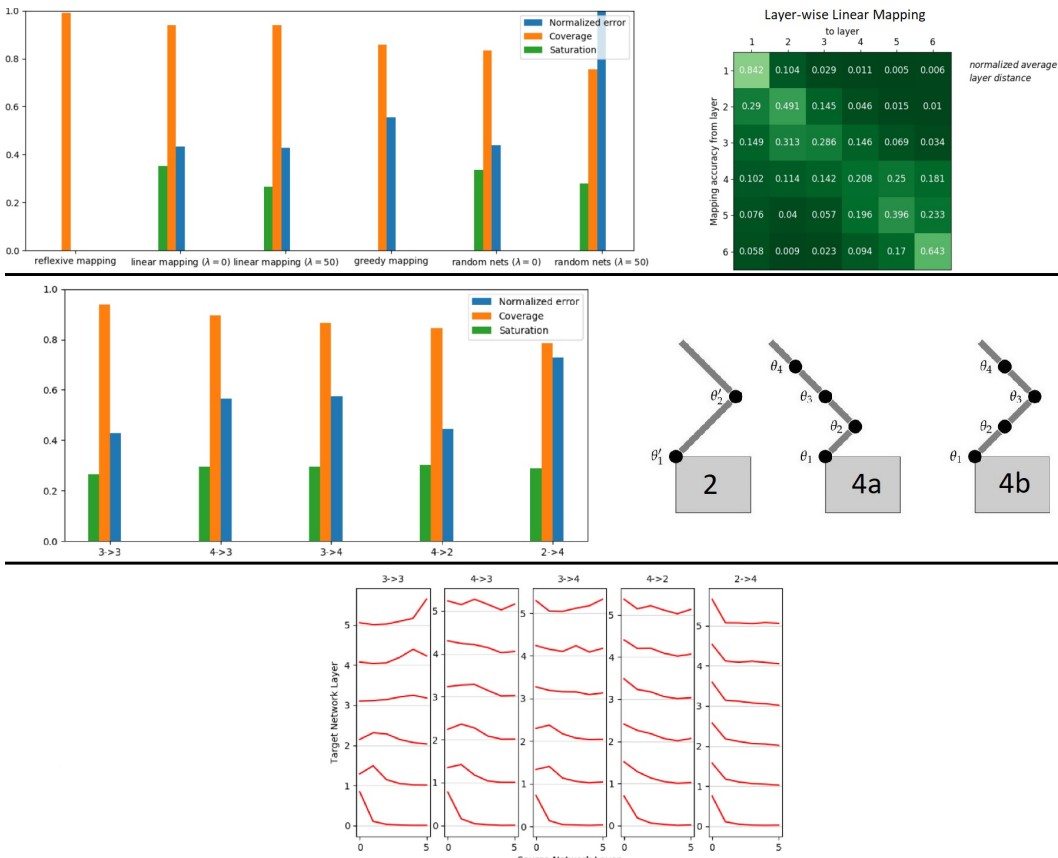

Figure 5: Projection of neuron activation between networks trained for variable joint robot manipulators.(data averaged with 5 networks each, 3 pruning steps). **Top:** Benchmark of mapping technique and evaluation metrics: On the example of multiple three joint manipulator networks, we find linear mapping with $\lambda = 50$ the superior mapping approach in contrast to greedy mapping. In particular, coverage and normalized error indicate mapping quality well in comparison to untrained networks. A layer-wise linear mapping with $\lambda = 50$ is not optimal, but strongest correlations can be found between corresponding layers. This is represented in the higher diagonal values in the table of normalized average layer distances on the right. Here, layer 1 and 6 show best mapping. (initialization with 6 layers each 256 neurons before pruning, random nets with average pruned network size of $s = [46 \quad 22 \quad 16 \quad 13 \quad 8 \quad 20]$ neurons per layer). **Middle:** Neuron activation correlations of networks trained on robots with different joint count (2-4 refers to a mapping from networks trained for a 2 joint to a 4 joint robotic arm): The mapping error gets higher with increased difference in joint numbers, the coverage accordingly decreases. Mapping a network with higher complexity into lower complex ones performs slightly better than vice versa. In this study the mappings 4-2 are closest to the performance of the native 3-3 mappings. This mapping is influenced by a proper transformation of sensory inputs to the increased number of input neurons on the first layer. The results are demonstrated for balanced mapping as $\theta'_1 = \theta_1, \theta'_2 = \theta_3, \theta_2 = 0, \theta_4 = 0$ (4b) which performs better than in contrast to naive mapping $\theta'_1 = \theta_1, \theta'_2 = \theta_2, \theta_3 = 0, \theta_4 = 0$ (4a). (results as mean of 25 mappings). **Bottom:** Mean layer to layer projections: Networks trained for more similar robots show better layer to layer mappings. The first layer of the source network shows high utilization for mappings to all other layers, the penultimate layer is the most unlikely to be utilized. Middle layers map reasonably well for 4-3 and 3-4 mappings, more distinct robots as 4-2 and strongest in 2-4 mostly utilize first layer neurons only.

six demonstrate the strongest correlation potentially due to increasingly specialized neurons at the input and output of the network.

Linear mapping ($\lambda = 50$) has been applied between sets of 2, 3 and 4 joint robot manipulators (Figure 5 middle left), Random networks are initialized by the average network size of the respective joint count as evaluated with pruning. Scenarios 3-4 and 4-3 show similar prediction errors but indicate a higher mean error compared to 4-2 mappings. Latter mapping performs similar to the baseline, which might be induced by the fact that we transform inputs in a balanced way so that the 4 joint arm can act like a 2 joint arm (figure on the right, we choose the transformation 4b). It shows lower coverage of the source network, which is partially related to the fixed input channels for the source networks after input transformation. The worst performance according to the prediction error is shown by scenario 2-4 as the two joint manipulator networks are barely able to replicate the behavior of the four joint networks. Generally, the more distinct the robots the worse the mapping, except input transformation is implemented in a meaningful way. More complex networks map slightly better into less complex one, as compared to the opposite way round.

A deeper insight to the source network utilization is drawn from mean layer-wise projections (Figure 5 bottom). The baseline scenario 3-3 shows more significant correlation to its respective layer the closer it is to the input or output. The first layers of 3-4 and 4-3 mappings seem to follow the behavior of the baseline, whereas the deeper layers show no significant correlation. Contrary to the performance of the overall metrics, scenario 4-2 shows no strong layerwise correlation, which is even worse in the inverted 2-4 mapping. If layers do not map well, all target layers tend to map to the lower layers especially the first layer (most prominent in 2-4 mappings) of the source network, only a small tendency is visible of the output layer mapping to other output layers. We hypothesize this phenomena is credited to first layers having the highest neuron count and activation variance. Overall, we do find that a good mapping correlation when the source network is able to imitate the behavior of the target network, a suitable input transformation turned out to be crucial here. 4-2 mappings showed the lowest error, but networks trained on three and four joint networks map better into their respective layer.

## 7 CONCLUSION

In this paper we analyzed individual neuron activation and correlations between neural networks trained for goal reaching of vertical space robot manipulators with two, three and four joints. We analyzed and classified the activation in order to implement a pruning algorithm that removes redundant neurons and increases information density in the network. Finally, we analyzed correlations between the overall and layerwise neuron activation of networks trained on robots with different joint number by projection mapping. Our results demonstrate that networks develop distinct activation patterns on individual neuron layers with bell-shaped distribution of activation densities. This distribution is compressed by our pruning algorithm that merges similar neuron activation classes mostly on the inner network layers. Networks trained for robots with only small joint number difference show a good correlation of neuron activation, for small differences this correlation can be found layer-wise. The more distinct the robot kinematic is in terms of joint number, the more important is a proper input transformation that fits the different network input layers. All experiments are benchmarked by comparison against untrained networks and self-correlations for multiple networks trained for the same task. Our results help to improve explainability of reinforcement learning in neural networks for robot motion learning and highlight network structures that can be reused on similar tasks after pre-training. The experiments conducted are limited to robot manipulators of 2 to 4 joints acting in vertical space, however the underlying introduced methodologies could be transferred to other Reinforcement Learning tasks as well. Analysis of neuron activation has been introduced in other contexts, here here we utilize it for the analysis of the specific use case of vertical space robot manipulation. In future work our pruning algorithm can be extended to also reduce the number of overall layers, analyze additional network parameters and we will examine reusing network structures with good correlation experimentally.

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
