# OpenReview forum: "Neuron Activation Analysis for Multi-Joint Robot Reinforcement Learning"
_ICLR.cc/2021/Conference — Reject_

### Official Review · AnonReviewer4 · 2020-10-26
**Interesting idea worth exploring, however, it needs to be developed further.**

**Rating:** 5
**Confidence:** 4

**Review:**

###  Summary

The authors investigate individual neuron activations over time, and compare the neuron activations within individual networks all-to-all and layer wise.

A distance metric is introduced and utilized to set up a pruning procedure to maximize the information density in learned neural networks and reduce redundancy as well as unused network nodes.

Finally, neuron activations are used to assess the correlations between learned policy networks for manipulators with a varying number of degrees of freedom. A projection mapping between different policy networks is implemented and analysed, as a type of transfer learning between different robot morphologies.


###  Review

I believe that this is an interesting work which tries to understand the inner-workings of a robot-control policy network by examining the network activations, and further using this information to prune the unnecessary neurons. Transferring learned network policies between robot morphologies is very useful and preliminary insights seem interesting.

There are some important flaws that need to be addressed regarding the clarity of the methodology and contributions, as well as the significance of the experimental evaluations.

My impression is that although the work tackles an important problem with a good idea, this is still an incomplete work, as the presented experimental evaluation is insufficient to draw significant conclusions.

Below are some of the comments organised by sections, including concrete suggestions for improving the work.

1. INTRODUCTION

 The main motivation and goal should be presented explicitly in a separate paragraph. There seems to be a missing link between the neuron activation estimations for pruning and correlation analysis for transfer mapping.

2. RELATED WORK

 I do not see the relevance of Bellemare et al. (2013), Mnih et al. (2015), Chess Silver et al. (2017), Go Silver et al. (2016) and Lillicrap et al. (2015) to the specific problems investigated in this paper.

 It would be useful to consider other work investigating NN complexity, and adding a discussion on how it relates to this work:
 - Gaier, Adam, and David Ha. "Weight agnostic neural networks." Advances in Neural Information Processing Systems. 2019.
 - Li, Chunyuan, et al. "Measuring the intrinsic dimension of objective landscapes." International Conference on Learning Representations. 2018.

3. EXPERIMENTAL SETUP

 “planar space robot manipulator that represents a multitude of real world applications” Do you mean that this task is a surrogate for examining many applications? This seems like a strong statement as it represents a small subset of potential applications.

 Moreover, planar space usually refers to having a horizontal plane as a task space. In this case it would be more clear to say “operating in a vertical plane”.

 “A neural network is trained with end-to-end Reinforcement Learning” this usually means from input images to output torques, but in the presented approach position control is used, so this should be emphasised.

 “physical robotic simulation”, usually it is said “physical robot” referring to a real robot experiment, or a “realistic robot simulation” which refers to a simulation that takes into account the real robot component values (dimensions, mass, inertia…).

 Equation 1 is not clear as the text says that $\textbf{x}$ consists of joint angles $\hat{\theta}_i$ but eq 1 shows the sin and cos projections of the angles. Moreover, the index $i$ seems to refer to both the time-step index $t_i$, as well as the joint index $\hat{\theta}_i$. Also, $n$ is not introduced as the number of joints.

 What is the reason behind mapping the target distance into a (0,1] range?

 Another important aspect which I believe should be addressed, is what is the effect of the task which is learned on the activation correlations? Basically, different tasks would have a different state distribution seen at the input of the policy? One simple example would be examining different types of control - position vs velocity vs torque.


4. NEURON ACTIVATION ANALYSIS

 This section provides an analysis for the 3DOF robot only and should be emphasised.

 “We define a distance metric between neurons that is based on the neuron activation history in scope of every episode in order to account for the dynamics in motion trajectory learning.” It is a bit unclear how the activation history is evaluated. This is one of the most important parts of the paper and should be made very clear.

 I am not 100% sure that the distance metric should be referred to as Euclidean distance, as it does not operate on euclidean space, so I think it would be sufficient to say “a proposed neural activation distance metric”.

 “For a set of sample episodes E representing the agents action space” How do episodes represent the action space exactly, I do not understand this part completely.

 The notation is a bit confusing, as $n$ refers both to the number of joints and the number of neurons. Also, what does the superscript $T$ in $R^T$ stand for?

 It seems to me that there is another summation missing in equation 2, as the indices of neurons should be more generic like $n_i$ and $n_j$ (unless there are only 2 neurons?). Also, are these neurons form the same layer or all the neurons in the network? This should be clarified in the text and formulated in Eq 2.

 What is $C$ in equation 3? I assume it is the cluster and cluster size, but this should be explained explicitly.

 Some of the distributions in Fig 2 are a bit skewed and look more like Beta distribution rather than Gaussian. Why is having a Gaussian distribution of distances relevant?

 There might be some other visualisation method that could be used here to shed light on the findings. Because currently, it is difficult to see any significant differences between the plots.

 The reference to Fig 2 should be improved, the definition of the trained, random and pruned lines is missing (both in the figure and the text).

 What does “all-to-all distribution of trained networks” mean (all-to-all neuron comparison)?

 The findings for the clustering (Fig 2 bottom) are very interesting! Could you maybe elaborate on these more?


5. HEURISTIC NETWORK PRUNING

 What is the motivation for retraining after pruning? If neurons that have similar activations are pruned, what would happen if one of them is kept with the corresponding weights? How would this affect the performance? This could be an insightful baseline comparison.

 Moreover, what is the advantage of reusing the network weights for initialisation, instead of randomly initialising them? This would also be an interesting experiment to conduct.

 Please introduce what are “dead neurons”.

 Wrong reference to Equation (5) should be (4)

 $\tau > 2$ → $\tau > 0.2$

 The accuracy in Fig 3 left, is given in [%], is this a mistake, because then it seems that the initial accuracy is only 0.8%. If this is actually 80%, why is the optimal $\tau = 0.2$ as the corresponding accuracy has fallen to 20%. How is this evaluated? Does the green label correspond to pruned or initial network size?


6. CORRELATIONS IN NETWORKS TRAINED FOR MULTI-JOINT ROBOTS

 It would be useful to start with a high level overview of what is the goal of finding these correlations in addition to how they are calculated. It seems that the correlations between networks are not examined, rather the mappings between them. Therefore this is slightly misrepresentative of what is actually being done.

 How are activation matrices A and B related to P? Also, what are $\alpha_m$ and $\beta_k$ ?

 What is the motivation behind making $\bar{P}^g_{km}$ sparse according to minimal distance in Greedy mapping, or applying L1 regularisation in Linear mapping? I assume your goal is to map joints 1-1 rather than combining them? Please explain this in a more clear way if possible.

 Equation 5 does not show the variable which is optimised. What does $\alpha^{\downarrow}_m$ stand for?

 Equations 6 and 7 are not referred to in the text.

 The quantities defined in equations 6, 7, 8 and 9 should be properly introduced as evaluation metrics and named accordingly, in a separate paragraph.

 Figure 5 correlations (top right) would be more impactful if represented with a heatmap matrix in addition to the numbers. The graphs on the bottom are not very clear.

 The discussion of the mapping results is very interesting and should emphasise the mapping between different robot morphologies. For example the difference between 4 -> 2 and 2 -> 4, where the latter has a higher error which could be expected as there is not enough information stored which can be decoded. Having additional comparisons of robot manipulators with larger differences in DOF would probably emphasise this and support the given conclusions better. This would also strengthen the paper significantly.

 Another metric which would be necessary to evaluate the transfer procedure, is to evaluate the mapped network on a test set of the reaching task.


OTHER COMMENTS:
 - Figure captions should be larger
 - Several typos
 - Consistency in using “3 joint manipulation task” or “3-DOF manipulator”
 - In the conclusion it is stated: “In this paper we analyzed individual neuron activation and correlations between neural networks trained for goal reaching of a variety of planar space robot manipulators.” Having the same robot structure with 3 different DOFs is not sufficient to be considered as a variety of manipulators.
 - It would significantly help the clarity of the paper to split certain sections into thematic paragraphs.

---

### Official Review · AnonReviewer3 · 2020-10-27
**Potentially a good paper requiring more conclusive results**

**Rating:** 4
**Confidence:** 4

**Review:**

# Summary
The paper presents a technique to compare networks trained to solve similar tasks trained in different context. The considered task is reaching with a robotic planar arm; the considered context is varied varying the robot degrees of freedom. The goal of the paper is to find correlations across neural activity patterns across networks trained to solve the same task in different contexts.

To achieve their goals, authors propose an heuristic network pruning algorithm to reduce the network size while keeping performance in training and evaluation. To correlate different networks, authors propose a technique to project a source network onto a target network.
#### Clarity
The paper is well written and easy to read.
#### Originality
The paper follows a main stream research which aims at pre-training neural networks on general task to speed up learning of specific tasks. As far as I know (I am not an expert in the field), the proposed analysis is original.
#### Significance
The significance of this work is relatively low. The results could be more conclusive with further analysis and experiments.
#### Major comments
* **Kinematic redundancy**. Authors have chosen a specific task, nominally reaching a target with a planar manipulator. Within this context, in presence of more than two degrees of freedom (i.e. with kinematic redundancy), the solution of the task itself is non-unique. Therefore fixing the context (i.e. fixing the robot kinematics, the robot geometry, etc) doesn't guarantee that the RL algorithm will find similar solutions across different runs. Actually, given the random training procedure each solution should come up with completely different strategy, exploiting the kinematic redundancy. If so, how do authors compare networks which exploit differently the kinematic redundancy? Are the proposed metrics (greedy mapping, linear mapping, etc) invariant with respect to different solutions to the same task (i.e. invariant to kinematic redundancies)?
* **Results and conclusions**. Goal of the paper (mentioned in the first two sentences of the abstract) is to progress in understanding if pre-training of end-to-end RL can be used as feature extractors and hierarchical organizations. Despite what claimed in the conclusions ("Networks trained for robots with only small joint number difference show a good correlation of neuron activation, for small differences this correlation can be found layer-wise."), authors fail short in giving a sound explanation of why this is the case.
#### Major comments
* **Page 7, line 8 from the top.** "[..] the reflexive mapping". This was not mentioned before, authors should give more details.
* **Page 7, line 1 from the bottom.** "Are joint numbers very different a proper input transformation is crucial to find correlations". Please check this sentence.
* **Page 8, caption of figure 5.** "balanced mapping θ1′ = θ1 , θ2′ = θ3 (4b) we apply in contrast to the naive mapping θ2′ =θ2,θ2′ =θ2(4a)." It's unclear what these mappings refer to and how they have been used.

---

### Official Review · AnonReviewer1 · 2020-10-27
**Thorough analysis in a limited scope**

**Rating:** 5
**Confidence:** 3

**Review:**

#### Summary
The authors present a method for analysing neuron activity in neural networks trained via RL on a multi-joint planar reaching task, as well as correlating neurons between different models trained on tasks with potentially a different number of joints. The methods consists of three steps:
1. Compare different neurons within a model using normalised activation traces over a number of episodes, and cluster hierarchically based on similar activity.
2. Use said clusters to prune the networks based on merging neurons within a cluster with an intra-cluster distance below some threshold, alternated with retraining.
3. Compare different models by optimising a linear projection between neurons, and evaluate reconstruction error, coverage and saturation.
Results indicate the proposed pruning method is effective in reducing the number of neurons without affecting accuracy, as well as showing correlations between corresponding layers of different models, though these reduce with larger difference in number of joints.

#### Pros
- The authors perform a sufficiently thorough evaluation, with a large number of models compared and reasonable ablations, baselines and metrics.
- While descriptions are brief, the method is generally well described and mathematical notation consistent.
- The proposed heuristic pruning approach seems to perform well in this case, as evident by all model sizes converging to the same size in Fig. 3.
- The approach of first pruning networks to maximise information content in the activations before correlating different models makes a lot of sense.

#### Cons
- The authors frame their work within the context of feature reuse and explainability, however the presented work is limited to showing correlations between features learned on identical or very similar tasks. It is unclear how this enables either reuse or explainability and perhaps not surprising that these correlations for very similar or identical tasks exists per se, more interesting would be to see how these can exploited. These correlations also degrade very rapidly with an increasing number of joints. I hypothesise that one would perhaps see stronger correlations between more different tasks if not the morphology was changed but rather the objective / reward. The scope of a planar reacher may also be too limited to draw more general conclusions for other control tasks.
- Related to task differentiation, a potential weakness in the proposed methods is how the activation traces are generated for source and target model when optimising the projection. Effectively only the target model is evaluated within distribution, after which the inputs observed there are then remapped to the source model. It's unclear what the effect is of potentially evaluating the source model out of its training distribution. I was hoping to see a way to correlate trajectories collected with the respective models independently. Perhaps the type of problem considered only allows a singular solution even across different number of joints, but it would be good to verify this.
- While the authors do evaluate a large combination of models, only averages are reported. Given how close the results seem to be to random in Fig. 5, it's hard to gauge the significance of the results. Some variance or error metric would be very valuable.
- While the idea of pruning the networks before correlating intuitively seems like a good idea, this is not experimentally validated. It would be good to add a comparison with and between unpruned models as well.
- While okay to follow, the text could use a bit more polish.

#### Questions
- There's currently no mention of how all these models were trained. One caption hints at DQN? Please provide more details.
- It's unclear how to interpret training duration in Fig. 3. Is this the time required to "pass" the validation set again after pruning?
- What do the values in the table in Fig. 5 represent? Sums of weights in the projection matrix?

#### Conclusion
While overall the method presented makes sense, and the evaluation is relatively thorough, the scope of the problems evaluated is considerably limited to draw any general conclusions of its validity, and some of the framing and details raise questions. As such I'd consider this submission marginally below acceptance.

---

### Decision · Program_Chairs · 2021-01-07
**Final Decision**

**Decision:**

Reject

**Comment:**

The paper analyzes neuron activations for neural networks trained via RL to perform reaching with planar robot arms. This analysis includes an evaluation of the correlation between neurons of different models trained to control arms with different degrees-of-freedom. In performing these evaluations, the paper proposes a heuristic pruning algorithm that reduces the size of the network and increases information density. Correlation is assessed based on a projection of the source network on the target network.

The paper is well written and considers a challenging problem of interest to the community. The proposed pruning strategy as a means of maximizing information content is reasonable and seems to perform well. However, the significance of the contributions is limited by the experimental evaluation. The experiments consider a large number of models, however the scope of problems on which the method is evaluated is narrow, making it difficult to draw conclusions about the merits and significance of the work. The authors are encouraged to extend the analysis to a more diverse set of problems.